# Discovery of a Novel Member of the *Carlavirus* Genus from Soybean (*Glycine max* L. Merr.)

**DOI:** 10.3390/pathogens10020223

**Published:** 2021-02-18

**Authors:** Thanuja Thekke-Veetil, Nancy K. McCoppin, Houston A. Hobbs, Glen L. Hartman, Kris N. Lambert, Hyoun-Sub Lim, Leslie. L. Domier

**Affiliations:** 1Department of Crop Sciences, University of Illinois, Urbana, IL 61801, USA; tthekke2@illinois.edu (T.T.-V.); houston.a.hobbs@gmail.com (H.A.H.); glen.hartman@usda.gov (G.L.H.); knlambert@illinois.edu (K.N.L.); 2Soybean/Maize Germplasm, Pathology, and Genetics Research Unit, United States Department of Agriculture-Agricultural Research Service, Urbana, IL 61801, USA; nancy.mccoppin@usda.gov; 3Department of Applied Biology, College of Agriculture and Life Sciences, Chungnam National University, Daejeon 305-764, Korea; hyounlim@cnu.ac.kr

**Keywords:** *Carlavirus*, *Betaflexiviridae*, soybean carlavirus 1, high throughput sequencing

## Abstract

A novel member of the *Carlavirus* genus, provisionally named soybean carlavirus 1 (SCV1), was discovered by RNA-seq analysis of randomly collected soybean leaves in Illinois, USA. The SCV1 genome contains six open reading frames that encode a viral replicase, triple gene block proteins, a coat protein (CP) and a nucleic acid binding protein. The proteins showed highest amino acid sequence identities with the corresponding proteins of red clover carlavirus A (RCCVA). The predicted amino acid sequence of the SCV1 replicase was only 60.6% identical with the replicase of RCCVA, which is below the demarcation criteria for a new species in the family *Betaflexiviridae*. The predicted replicase and CP amino acid sequences of four SCV1 isolates grouped phylogenetically with those of members of the *Carlavirus* genus in the family *Betaflexiviridae*. The features of the encoded proteins, low nucleotide and amino acid sequence identities of the replicase with the closest member, and the phylogenetic grouping suggest SCV1 is a new member of the *Carlavirus* genus.

## 1. Introduction

Soybean is one of the most important cash crops in the United States, where it is cultivated on more than 30 million hectares annually [1]. The crop is affected by many pathogens, including a growing number of viruses, of which bean pod mottle virus and soybean vein necrosis orthotospovirus have the highest incidence in most years in North America [2,3]. 

High-throughput sequencing (HTS) has led to a significant increase in the discovery of novel plant viruses. The goal of this study was to monitor soybean production fields in the state of Illinois, USA for the relative abundance of previously described and emerging viral pathogens by HTS. From RNA-seq analysis of total RNA from soybean leaves, we discovered a putative new member of the *Carlavirus* genus, tentatively named soybean carlavirus 1 (SCV1). The genus *Carlavirus* is one of 13 genera in the subfamily *Quinvirinae* and the family *Betaflexiviridae* (order *Tymovirales*) [4]. Carlaviruses have single-stranded positive-sense RNA genomes of 8.3–8.7 kb with a 5’-m7 G cap and a 3’ poly(A) tract. Carlavirus genomes typically contain six open reading frames (ORFs) encoding, in order, replicase, triple gene block proteins (TGBps), coat protein (CP) and nucleic acid-binding protein (NABP) [5]. This communication describes the molecular features of SCV1 and its phylogenetic relationship with other recognized members of the family *Betaflexiviridae*.

## 2. Results

### 2.1. High-Throughput Sequencing

Near full-length sequences for SCV1 were assembled from RNA-seq data from soybean leaf samples collected in 2008 (MW349427; 207,869 reads), 2009 (MW176107; 41,034 reads) and 2015 (MW349428; 6351 reads). In 2010 and 2012–2014, less than 50 reads were detected for SCV1 in the collected field samples. Even though thrips are not reported to vector carlaviruses [4], we also assembled a SCV1 genome (MT293130) from RNA-seq data of soybean thrips (*Neohydatothrips variabilis*, Beach) that were captured in 2018 by the Midwest Suction Trap Network [6]. 

### 2.2. Soybean Carlavirus 1 Genome Organization

The complete genome obtained for the 2009 SCV1 isolate is 8681 nucleotides (nt) long excluding the poly(A) tract with a genome organization typical of carlaviruses (Figure 1A). Despite the use of different copy number cloning vectors, different growth conditions and different *Escherichia coli* strains, attempts to construct infectious full-length clones of SCV1 were unsuccessful because the cloned virus genomes were unstable in *E. coli*. 

The pairwise genome sequence identities of the 2009 isolate with those of the 2008, 2015 and 2018 SCV1 isolates ranged from 93% to 98% compared to less than 63% for the genome sequences of red clover carlavirus A (RCCVA) isolates (Figure 1B). The 5’ untranslated region (UTR) and the 3’ UTR are 82 nt and 80 nt, respectively. The genome contains six ORFs encoding the replicase polyprotein, TGBps, CP and NABP. ORF1 (nt 83–6136) and ORF2 (nt 6171–6863) are separated by 34 nt. ORF2 overlaps ORF3 (nt 6838–7161) by 26 nt. ORF3 and ORF4 (nt 7167–7352) are separated by five nt. ORF4 is separated from ORF5 (nt 7394–8293) by 41 nt, and ORF5 overlaps ORF6 (nt 8290–8601) by four nt. In carlaviruses, only ORF1 is translated from genomic RNA, the product of which is autocatalytically processed by the encoded papain-like proteinase [7]. Proteins downstream are translated from 3’ co-terminal subgenomic RNAs [4]. 

ORF1 encodes a 227 kDa replicase polyprotein that contains domains for methyltransferase (pfam01660; amino acid (aa) Tyr_44_- Glu_356_), an AlkB member of the 2-oxoglutarate- and Fe(II)-dependent oxygenase superfamily (pfam13532; aa Cys_706_- Phe_812_), carlavirus endopeptidase (pfam05379; aa Lys_1031_- Tyr_1,118_), viral RNA helicase (HEL) (pfam01443; aa Gly_1,212_- Asn_1,474_) and RNA-dependent RNA polymerase (RdR) (pfam00978; aa Met_1,597_- Asn_2,008_). The ORF1-encoded protein was most similar to the corresponding protein from red clover carlavirus A (RCCVA; GenBank accession, KY474546.1) with 60.6% aa sequence identity. 

ORF2 is predicted to encode a 26 kDa TGBp1 that contains a HEL domain (pfam01443; aa Val_24_- Cys_221_). ORF3 putatively encodes an 11 kDa TGBp2 that contains a domain (pfam01307; aa Thr_4_- Cys_103_) that is present in movement proteins of carlaviruses, hordeiviruses and potexviruses. ORF4 is predicted to encode a 7 kDa TGBp3 that contains a pfam02495 domain that is commonly present in TGBp3s of carlaviruses and potexviruses. The TGBps of SCV1 were most similar to the corresponding proteins of RCCVA with 64.9%, 75.7% and 55.9% aa sequence identities, respectively, for TGBp1, TGBp2 and TGBp3. ORF5 is predicted to encode the 35 kDa CP that contains two CP conserved domains (pfam08358 (Arg_51_- Leu_102_) and pfam00286 (Ser_111_- Glu_250_)) typically found in carlaviruses. Like other carlaviruses, ORF6 of SCV1 was predicted to encode a 12 kDa protein with a nucleic acid-binding domain (pfam01623; aa Arg_4_- Ile_93_) with a cysteine-rich motif and a centrally located arginine-rich nuclear localization signal. SCV1 CP and NABP showed the highest aa identities (83.6% and 74.8%, respectively) with the orthologues in RCCVA. The ORF6 proteins of chrysanthemum virus B and potato virus M were shown to be involved in nucleic acid binding [8,9]. Carlavirus NABPs, along with the TGBp1s, which function as suppressors of RNA silencing, have been reported to be pathogenicity determinants [10]. 

### 2.3. Pylogenetic Analyses

In phylogenetic analyses with members of the *Betaflexiviridae* using the predicted aa sequences of the replicase and CP, SCV1 consistently grouped with RCCVA, members of the *Carlavirus* genus and members of the subfamily *Quinvirinae* (Figure 2). The species molecular demarcation criteria for the family *Betaflexiviridae* are 72% nt identity (or 80% aa identity) in the replicase or CP genes [4]. Although SCV1 CP exhibited 83.6% aa identity with the CP of RCCVA, the replicase aa sequence identity of 60.6% (60.4% nt identity) is far below the demarcation criteria. Phylogenetic grouping of SCV1 with recognized *Carlavirus* members, molecular features of the encoded proteins and low aa identity of replicase below the demarcation criteria confirm that SCV1 is a novel species in the genus *Carlavirus*.

## 3. Discussion

Currently, the *Carlavirus* genus contains 53 species recognized by the International Committee on Virus Taxonomy [11]. The genus *Carlavirus* is in the subfamily *Quinvirinae*, members of which share five conserved proteins (replicase, three TGBps and CP) [4]. In contrast, members of the subfamily *Trivirinae* share only three conserved proteins (replicase, 30K movement protein and CP). The higher levels of sequence conservation among members of the *Quinvirinae* are evidenced by higher bootstrap values in the phylogenetic analysis of the selected aa sequences (Figure 2).

SCV1 is the second member of the *Carlavirus* genus identified naturally infecting soybean. The other member, cowpea mild mottle virus (CPMMV), is transmitted by whiteflies, causes severe symptoms in soybean in Africa, Asia and South America and has been detected in legumes and whiteflies in the USA [12,13]. In RNA-seq data from the soybean leaf samples from 2008–2015, SCV1 showed predominant occurrence in 2008 and 2009 but was not detected at significant levels again until 2015. The apparent differences in the incidence of SCV1 infection could be related to differences in the abundance of biological vectors that have not been identified yet. Other carlaviruses are transmitted by aphids or whiteflies in a nonpersistent or semipersistent manner and some are also spread by seed and mechanical contact [4]. The nucleotide sequence of the genome of the soybean thrips isolate of SCV1 identified in 2018 was 95.9% identical to the original isolate assembled from the soybean leaf RNA-seq data. Soybean thrips might have acquired the virus while feeding on soybeans or an alternative host.

Like many members of the *Betaflexiviridae*, some carlaviruses cause mild symptoms, or are symptomless [4,14]. Some other members of the genus, e.g., blueberry scorch virus, CPMMV and melon yellowing-associated virus, are capable of inducing significant crop losses [15,16,17]. The severity of symptoms induced by SCV1 and its epidemiology, including the identification of its biological vectors, need to be explored further.

## 4. Materials and Methods

### 4.1. High-Throughput Sequencing

Total RNAs were extracted using RNeasy Mini Kits (QIAGEN, Valencia, CA, USA) from soybean leaves that had been randomly collected without regard for symptoms in 2008–2015 as part of field surveys conducted in Illinois, USA. Total RNAs were depleted of ribosomal RNA using a Ribo-Zero rRNA Removal kit (Epicentre Biotechnologies, Madison, WI, USA). Libraries were prepared using TruSeq Stranded Total RNA Sample Prep kits (Illumina, Inc., San Diego, CA, USA), and sequenced on an Illumina HiSeq 2000 at the W. M. Keck Center for Comparative and Functional Genomics at the University of Illinois, USA. Single- and paired-end sequence reads (100 nt in length) were assembled using the Trinity *de novo* transcriptome assembler [18], and compared to available viral aa and nt sequences using BLAST searches [19]. 

### 4.2. Random Amplification of cDNA ends (RACE)

To confirm the terminal sequences of the 2009 SCV1 genome, RACE was performed on total RNA extracted from RT-PCR-positive leaf samples using the First-Choice RLM-RACE kit (ThermoFisher, Waltham, MA, USA). Inner and outer primers used to obtain the 5’ terminus were 5’-CGATAACTTCTTTTTCGCCTCTGCCTTC-3’ (nt 226–253) and 5’-TGGAAGCTATGGCGGACTGTACGCTGG-3’ (nt 138–164), respectively. Inner and outer primers used to obtain the 3’ terminus were 5’-CCACCGTTCTACTTTACCACAAGGTGTGAT-3’ (nt 8485–8514) and 5’-CATTCAATTCGGAGTAACTGAGGTGATACC-3’ (nt 8565–8594), respectively. Amplicons were sequenced directly using the BigDye Terminator Cycle Sequencing Kit (ThermoFisher) at the Roy J. Carver Biotechnology Center at the University of Illinois. 

### 4.3. Construction of Full-Length Clones of SCV1

Full-length cDNAs were amplified from soybean total RNA using iProof high-fidelity DNA polymerase (BIORAD, Hercules, CA, USA) with virus-specific primers (5’-TAAAGAAAAACACACACACCAAACATACACAAAC-3’ and 5’-TACGGTATCGA(T)_30_AAGTAAAAATAGTTAAAAAC-‘3). For direct DNA inoculations, amplicons were cleaved with *Cla*I and cloned downstream of the cauliflower mosaic virus (CaMV) 35S promoter in pBr35S [20] with and without a hepatitis delta virus antigenomic ribozyme [21] between the poly(A) tail and the NOS terminator. The amplicon also was cloned into pHST40 [22] downstream of the CaMV 35S promoter. For inoculation with in vitro transcripts, full-length cDNAs of the SCV1 genome were amplified with a 5’ primer containing a T7 RNA polymerase promoter sequence (5’-TATCGGCAATAATACGACTCACTATAGGGTAAAGAAAAACACACACACCAAACATACACAAAC-3’) and the 3’ virus-specific primer. The amplicon was cloned into the pCR-BluntII cloning vector (ThermoFisher). In attempts to stabilize the plasmids, *E. coli* transformed with the plasmids were grown at 28 °C instead of 37 °C. To reduce possible negative effects of toxic viral proteins, ligation reactions also were transformed into ABLE C *E. coli* cells [23]. 

### 4.4. Genome comparison

The genome organization of SCV1 was obtained using National Center for Biotechnology Information ORF Finder (www.ncbi.nlm.nih.gov/orffinder). For sliding window comparisons, nucleotide sequences of 2008, 2009, 2015 and 2018 isolates of SCV1 and the B46 (KY474546.1) and Manza 2014 (KY471462.1) isolates of RCCVA were aligned using Clustal Omega [16]. Percent nucleotide sequence identities with the 2009 SCV1 sequence were calculated and plotted for 100 nt windows using Microsoft Excel.

### 4.5. Phylogenetic Analyses

For phylogenetic analyses, predicted aa sequences of replicase and CP were aligned with the sequences of selected recognized members of the *Betaflexiviridae* by Clustal Omega. Phylogenetic trees were constructed using the maximum likelihood method in MEGA X [24] with 500 bootstrap replications.

## Figures and Tables

**Figure 1 pathogens-10-00223-f001:**
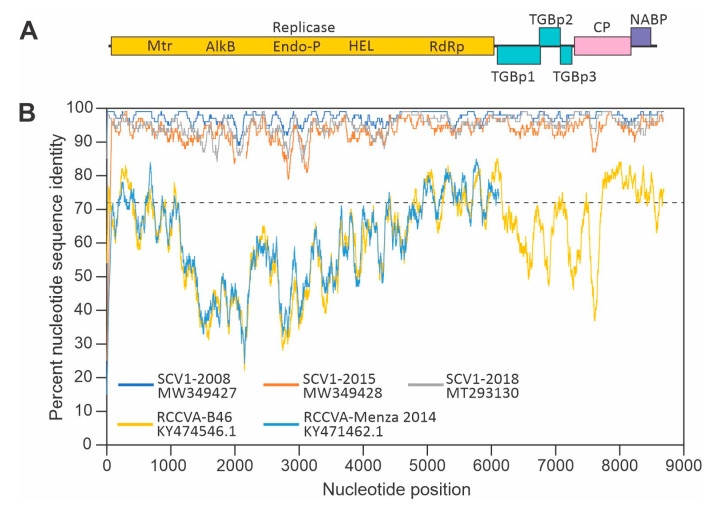
(**A**) Schematic representation of the soybean carlavirus 1 (SCV1) genome. The predicted replicase amino acid sequence contains methyltransferase (Mtr), AlkB 2OG-Fe(II) oxygenase, endopeptidase (Endo-P), helicase (HEL) and RNA-dependent RNA polymerase (RdRp) domains. Downstream, the genome encodes triple gene block proteins (TGBp) 1–3, a coat protein (CP) and a nucleic acid-binding protein (NABP). (**B**) Sliding window (100 nt) comparisons of nucleotide sequence identities between the genome of the 2009 isolate of SCV1 and genome sequences of 2008, 2015 and 2018 isolates of SCV1 and two isolates of red clover carlavirus A (RCCVA). The dashed line indicates the 72% nucleotide sequence identity threshold for species demarcation within the family *Betaflexiviridae*.

**Figure 2 pathogens-10-00223-f002:**
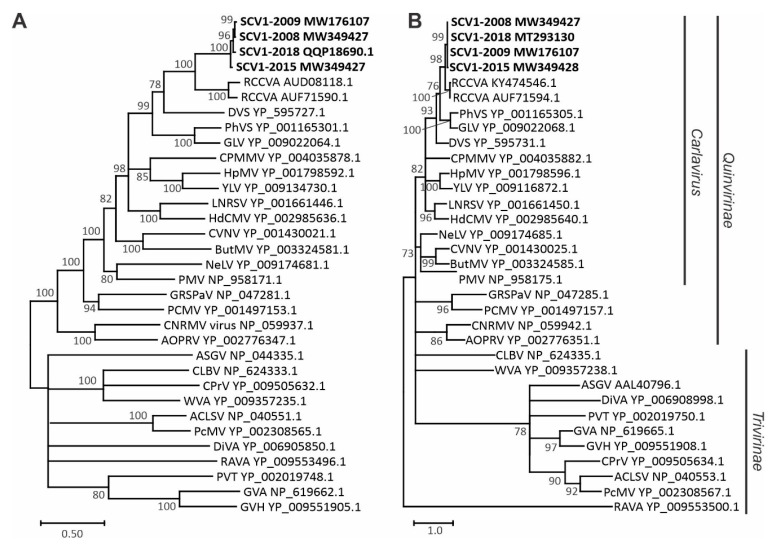
Phylogenetic relationships of soybean carlavirus 1 with selected members of the family *Betaflexiviridae* based on replicase (**A**) and coat protein (**B**) predicted amino acid sequences. Nodes with less than 70% bootstrap support were collapsed to the next higher level. Virus sequences included in the analysis were African oil palm ringspot virus (AOPRV), apple chlorotic leaf spot virus (ACLSV), apple stem grooving virus (ASGV), butterbur mosaic virus (ButMV), Caucasus prunus virus (CPrV), cherry necrotic rusty mottle virus (CNRMV), citrus leaf blotch virus (CLBV), coleus vein necrosis virus (CVNV), cowpea mild mottle virus (CPMMV), daphne virus S (DVS), Diuris virus A (DiVA), gaillardia latent virus (GLV), grapevine rupestris stem pitting-associated virus (GRSPaV), grapevine virus A (GVA), grapevine virus H (GVH), hop mosaic virus (HpMV), hydrangea chlorotic mottle virus (HdCMV), ligustrum necrotic ringspot virus (LNRSV), nerine latent virus (NeLV), peach chlorotic mottle virus (PCMV), peach mosaic virus (PcMV), phlox virus S (PhlVS), poplar mosaic virus (PMV), potato virus T (PVT), red clover carlavirus A (RCCVA), Ribes americanum virus A (RAVA), soybean carlavirus 1 (SCV1), watermelon virus A (WVA) and yam latent virus (YLV).

## Data Availability

All sequences have been deposited in GenBank at the accession numbers given in the text. HTS data are available in the National Center for Biotechnology Information Short Read Archive (SRA) under accession number PRJNA614937.

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
