# Peer review of "Discovery of a Novel Member of the Carlavirus Genus from Soybean (Glycine max L. Merr.)"

_pathogens, 2021, doi:10.3390/pathogens10020223_

Round 1
Reviewer 1 Report
Dear Authors,
I have a great honor to review manuscript entitled “Discovery and characterization of a novel member of the Carlavirus genus from soybean (Glycine max L. Merr.)” which is considered Pathogens journal. I analyzed whole work and it is present good and firm scientific results. I suggest only several improvements.
I strongly suggest to more firmly precise aim of the study in Introduction section we use sentence Aim of the study was… or something similar. I would also kindly suggest to enlarge the Figure 1 and especially Figure 2. The bigger figures will be great asset of of this publication. I would also suggest to check Reference list because it is not prepared according Journal rules.
Sincerely,
Author Response
As suggested by the reviewer, a sentence describing the goals of the study has been added to the introduction.
The sizes of the fonts in Figures 1 and 2 were increased as suggested.
The references were already formatted for Pathogens. No changes were made
Reviewer 2 Report
I am confused about the paper type of this manuscript.
Is really research article?
This manuscript is seem like to "short communcation" , but registration of type is "Article".
If it's really an article paper, there is not fully description about the research.

Author Response
The article type has been changed to “Communication” as suggested by the reviewer.
Reviewer 3 Report
This manuscript describes the identification of a new virus from soybean which, based on the complete genome was assigned to the genus carlavirus and from RDRP and CP phylogeny, further classified as being a distinct virus separate from the next relative RCCVA, another carlavirus from red clover. The paper is well written. However, the title implies characterization and this is not the case as there are no biological data, no transmission, no symptom description, and no other information provided except a complete genome assembled from NGS data only. An annotated sequence report may be a more proper format for this type of work. Furthermore, a deeper analysis incl. RDP would be necessary to state the taxonomic status of the new virus isolate and to discriminate from RCCVA.
Author Response
As suggested by the reviewer, the word “characterization” was removed from the title. It is not clear what type of deeper analysis the reviewer is referring to. The species demarcation criteria for the family Betaflexiviridae do not include descriptions of that type of analysis. We showed that isolates of soybean carlavirus 1 were consistently different from isolates of the most closely related virus, red clover carlavirus A, over a ten-year period, and that the differences exceeded the species demarcation criteria elaborated by the ICTV. This also was graphically illustrated in the sliding window comparisons of the genomes of SCV-1 and RCCVA isolates in Figure 1B.
Round 2
Reviewer 2 Report
I agree to accept for this paper